# Aquaporins Involvement in Pancreas Physiology and in Pancreatic Diseases

**DOI:** 10.3390/ijms20205052

**Published:** 2019-10-11

**Authors:** Tatjana Arsenijevic, Jason Perret, Jean-Luc Van Laethem, Christine Delporte

**Affiliations:** 1Laboratory of Experimental Gastroenterology, Université Libre de Bruxelles, 1070 Brussels, Belgium; Tatjana.Arsenijevic@erasme.ulb.ac.be (T.A.); jl.vanlaethem@erasme.ulb.ac.be (J.-L.V.L.); 2Department of Gastroenterology, Hepatology and Digestive Oncology, Hôpital Erasme, Université Libre de Bruxelles, 808, Route de Lennik, 1070 Brussels, Belgium; 3Laboratory of Pathophysiological and Nutritional Biochemistry, Université Libre de Bruxelles, 1070 Brussels, Belgium; jason.perret@ulb.ac.be

**Keywords:** aquaporins, pancreas, pancreatitis, cancer

## Abstract

Aquaporins are a family of transmembrane proteins permeable to water. In mammals, they are subdivided into classical aquaporins that are permeable to water; aquaglyceroporins that are permeable to water, glycerol and urea; peroxiporins that facilitate the diffusion of H_2_O_2_ through cell membranes; and so called unorthodox aquaporins. Aquaporins ensure important physiological functions in both exocrine and endocrine pancreas. Indeed, they are involved in pancreatic fluid secretion and insulin secretion. Modification of aquaporin expression and/or subcellular localization may be involved in the pathogenesis of pancreatic insufficiencies, diabetes and pancreatic cancer. Aquaporins may represent useful drug targets for the treatment of pathophysiological conditions affecting pancreatic function, and/or diagnostic/predictive biomarker for pancreatic cancer. This review summarizes the current knowledge related to the involvement of aquaporins in the pancreas physiology and physiopathology.

## 1. Introduction

Pancreas is a gland with endocrine and exocrine function and represents a key organ for overall body homeostasis. Indeed, endocrine pancreatic β-islets are responsible for insulin secretion in response to hyperglycemia, therefore contributing to glucose homeostasis [1]. Furthermore, exocrine pancreatic acinar and ductal cells ensure pancreatic fluid secretion composed of water, ions and enzymes involved in food digestion [2].

Various diseases can affect physiological functions or arise from pancreatic dysfunction. The diseases include diabetes [3,4], acute and chronic pancreatitis [5], cystic fibrosis [6], pancreatic cancers [7] and pancreatic insufficiency leading to malabsorption syndrome [8].

AQPs are expressed in both endocrine and exocrine pancreas and ensure important physiological functions related to insulin secretion and pancreatic fluid secretion [9,10,11,12,13,14].

Aquaporins (AQPs) are a family of transmembrane water channel proteins that are ubiquitously expressed among animals, plants and microorganism [15]. Mammalian AQPs are classified as classical AQPs permeable to water (AQP0, AQP1, AQP2, AQP4, AQP5, AQP6 and AQP8); aquaglyceroporins (AQP3, AQP7, AQP9, AQP10 and AQP11) permeable to small solutes such as glycerol and urea in addition to water [16]; and unorthodox AQPs (AQP11 and AQP12) with still uncertain permeability (Table 1) [17]. Another sub-group of AQPs has been described recently, named peroxiporins, consisting of aquaporins that facilitate the diffusion of H_2_O_2_ through cell membranes. Several AQPs belonging to classical or aquaglyceroporins have been described as peroxiporins, namely AQP1, AQP3, AQP5, AQP8, AQP9 and AQP11 [18,19,20,21,22]. However, recent reports suggest that all AQPs should exhibit H_2_O_2_ permeability, although at varying degrees [23].

AQPs are present in all living organisms. They have evolved from a commune ancestor by initial internal duplication and inversion of a transmembrane portion containing three helices, yielding the hexa-helical structure. Two critical transmembrane peptide motifs, asparagine–proline–arginine (NPA), evolved ensuring pore selectivity. Indeed, NPA motifs evolved as a cationic selectivity filter and an aromatic residue and arginine (ar/R) region as a proton selectivity filter. Modification of the ar/R portion, namely in transmembrane domain 5, resulted in variable pore restriction leading to subfamilies such as the more permissive aquaglyceroporins [24]. Whole chromosome duplications, local duplications, horizontal transfer, provided variable number of copies that evolved over time and speciation events [25]. Individual gene sequences changed, yielding structural and thus functional variants, through selective adaptive pressure, in response to environmental cues such as energy source availability and also internal physiological needs and constraints [26]. The latter has led over time to structural variations determining functional specificities in terms of permeating solutes, tissue distribution and specificity.

Consequently, in mammals and humans in particular, AQPs facilitate water transport to ensure the maintenance of water balance, facilitate glycerol transport to ensure glycerol metabolism, and participate to other cell processes including cellular migration, cellular expansion and cellular adhesion occurring notably in tumor cells [27,28]. Thereby, AQPs are expressed in a wide range of tissues and organs participating to fluid secretion and/or absorption (such as kidneys, lungs, eyes, central nervous system and exocrine glands including pancreas), in glycerol metabolism (such as adipose tissue, liver, heart and endocrine pancreas) and in a wide variety of cancer cells. Namely, AQPs are expressed in a pancreatic tumor as well [29,30].

From a clinical point of view, AQPs potentially represent new therapeutic targets for the treatment of diseases [31,32,33,34]. As a consequence, AQPs inhibitors and gene therapy using viral vectors encoding AQPs have been developed [31,33,34,35,36,37,38]. In addition, AQPs are potential biomarkers for the diagnosis and prognosis of diseases, and in particular of cancers [29,39,40,41,42,43,44,45,46,47,48,49,50].

This review summarizes the current knowledge on the expression and functions of AQPs in both endocrine and exocrine pancreas, as well as the current potential interest to use AQPs as new useful tools for diagnosis, prognosis and treatment for pancreatic-related diseases and pancreatic cancers. 

## 2. Endocrine Pancreas

### 2.1. Morphology and AQPs Distribution

Endocrine pancreas accounts for circa 10% of total pancreatic cells and is composed of β-islets containing around 60% β-cells producing insulin, 30% α-cells producing glucagon and 10% of other γ-cells, δ-cells (or PP-cells) and ε-cells producing respectively somatostatin, pancreatic polypeptide (PP) and ghrelin [51]. Human β-islets are made of all the endocrine cells types mentioned above distributed randomly, whereas rodent β-islets are made of β-cells surrounded by the other three types of endocrine cells [51].

Divergent results obtained using distinct methodologies suggested at first an uneven distribution of β-islets in human [52] and rodent [51,53] pancreas, and more recently an even distribution of β-islets in human pancreas [54]. However, the region of the pancreas that is removed during human pancreatectomy had distinct metabolic consequences [55]. Indeed, the removal of the pancreatic head led to improved tolerance to glucose while the removal of the pancreatic tail led to elevated glucose concentration after fasting or the glucose-tolerance test [55]. In light of these data, it was hypothesized that not all β-islets are alike [51]. This hypothesis was based on distinct developmental origins of the head and the tail of the pancreas, distinct metabolic behavior of the β-islets originating from these regions, and molecular heterogeneity. Indeed, the head arises from the dorsal and ventral pancreatic buds while the tail and the central portion arises from the ventral pancreatic bud [56]. In addition, rodent β-islets showed greater capacity to synthesize and secrete insulin when originating from the dorsal pancreatic bud as compared to when originating from the ventral pancreatic bud [57]. Furthermore, single-cell profiling of both the messenger RNA and protein levels revealed the existence of four distinct groups of β-cells with unique antigenic and molecular characteristics [58]. In the future, therapeutic interventions may have to take into account the heterogeneity of β-islets cellular composition.

To the best of our knowledge, AQP distribution remains unknown in human endocrine pancreas. However, the expression of several AQPs has been documented in mouse and rat β-cells, but to date, not in other β-islet cell types. Indeed, the expression of the aquaglyceroporins AQP7 was detected on mouse and rat β-cells [59,60,61,62], as well as rat β-cell lines BRIND-BD11 and RIN-m5F [63,64,65]. In addition, the expression of AQP5 and AQP8, two classical AQPs, was also detected on mouse β-cells, and AQP12 was detected in β-cells of rat Langerhans islets (Table 2) [60,65].

### 2.2. Physiology and AQPs Functions

Due to the sole documented expression of AQPs within β-cells, this section will then only focus on the physiology of β-cells and the involvement of AQPs in β-cells physiology.

β-cell physiology has been extensively studied owing to its capacity to synthetize and secrete insulin, playing a key role in the control of glucose homeostasis, but also in diabetes linked to either insulin insufficiency or resistance [51,66,67]. Briefly, insulin secretion is induced postprandially by an increase in blood glucose, free fatty acids and amino acids concentrations. In the current model of stimulus-secretion coupling, β-cells behave as electrically excitable fuel sensors to trigger insulin secretion in a biphasic manner in response to glucose stimulation [1,68]. Indeed, upon glucose entry in β-cells and its metabolization, subsequent events include ATP/ADP ratio increase, closure of ATP-sensitive potassium (K_ATP_) channels, depolarization of plasma membrane, opening of voltage-dependent calcium channels and sodium channels, increase in intracellular calcium concentration, activation of calcium-binding proteins and finally exocytosis of insulin-containing granules [1,68]. This stimulus-secretion coupling pathway leads to a first phase of insulin secretion occurring rapidly (within 5–10 minutes) and robustly (accounting for circa 15% of total insulin secretion within 1 h stimulation) following glucose stimulation due to the recruitment of insulin-containing granules localized in the close vicinity of the plasma membrane [68]. A second phase of insulin secretion, recruits insulin-containing granules from intracellular storage pools to the plasma membrane in a slower (within 1 h following glucose stimulation) but efficient manner (accounting for circa 85% of total insulin secretion) [68].

The glucose-elicited insulin secretion results from a triggering pathway (also previously referred as the K_ATP_-ATP-dependent pathway) and an amplifying pathway (also previously referred as the K_ATP_-ATP-independent pathway) [1,68]. The latter pathway depends on the initial triggering signal and increased sensitivity of insulin-containing granules to intracellular calcium concentration increase [1,68]. Although the triggering and amplifying pathway have been associated with the first and second phases of insulin secretion, respectively, the amplifying pathway takes part in the first phase as well [1,68].

In addition to its triggering function played in the stimulus-secretion coupling, increased glucose concentration has been shown to lead to an augmentation in β-cell volume [69]. An increase in β-cell volume is known to activate the volume-regulated anion channel (VRAC) and induce plasma membrane depolarization, which in turn stimulates voltage-sensitive calcium channels and leads to intracellular calcium concentration increase and insulin release [70,71].

While functional studies have been conducted to assess the role of AQP7 (an aquaglyceroporin) in β-cells, such studies have been lacking to assess the role of both AQP5 and AQP8 (classical AQPs) in these cells. Consequently, only the physiological role of AQP7 in β-cells will be primarily discussed hereafter.

*Aqp7* knockout mice displayed hyperinsulinemia [61,62] with either hyperglycemia [62] or normoglycemia [61], or normal insulin and glucose levels [60]. In addition, some *Aqp7* knockout mice are characterized by β-cells displaying reduced mass and size, decreased insulin content, increased basal and glucose-induced insulin secretions, increased glycerol and triglycerides contents, elevated glycerol kinase activity, and lowered glycerol release upon lipolysis stimulation [61]. However, other *Aqp7* knockout mice displayed decreased basal and glucose-stimulated insulin secretions [60] and did not corroborate basal and glucose-stimulated insulin secretions observed in a previous study [61]. The apparent discrepancies observed among different *Aqp7* knockout mice could result from distinct mice genetic background and/or methodologies.

In the light of AQP7 glycerol permeability, the effects of glycerol (acting as an osmolyte), entry and metabolism have been studied in β-cells. Isosmotic or hyperosmotic addition of glycerol to the extracellular medium of rat β-cells sequentially induced cell swelling, VRAC activation, plasma membrane depolarization, electrical activity and insulin exocytosis [59]. In contrast to the effects of urea (another molecule acting as an osmolyte that can be transported by AQP7), the effects of glycerol were maintained during the exposure of β-cells to osmolytes [70]. The glycerol-induced β-cell activation was supposed to result from both its cell entry and subsequent metabolization [70]. Incubation of rat β-cell BRIN-BD11 incubated with extracellular hypotonic medium or isotonic medium deprived of 50 mM NaCl but replaced with 100 mM urea induced both [2-^3^H]glycerol entry and insulin release, as compared to isotonic medium [63]. In addition, insulin released by BRIND-BD11 cells upon incubation with isotonic medium deprived of 50 mM NaCl but replaced with 100 mM urea or 100 mM glycerol was inhibited following VRAC inhibition [63]. All together these data showed that urea and glycerol entry upon extracellular isotonicity led to cell swelling, VRAC activation, and subsequent events leading finally to insulin release [63]. Furthermore, the role of AQP7 assessed using β-cells from *Aqp7*^+/+^ and *Aqp7*^-/-^ mice confirmed that glycerol entry occurred via AQP7 and induced subsequent cell swelling, VRAC activation and plasma membrane depolarization and insulin release [60]. In addition, modification of the *AQP7*^-/-^ β-cells response to glucose or extracellular hypoosmolarity suggested a direct or indirect role for AQP7 at a distal or downstream site in the stimulus-secretion coupling [60]. Additionally, in RIN-m5F β-cells, the incretin hormone glucagon-like peptide-1 (GLP-1) downregulates AQP7, with AQP7 expression being negatively associated with insulin release [65]. Interestingly, the orexigenic and lipogenic hormone ghrelin downregulates AQP7, leading to an increased cytosolic glycerol content that promotes triacylglycerol synthesis [65]. In this regard, it seems plausible that the reduction of AQP7 induced by GLP-1 and ghrelin might result in intracellular glycerol accumulation, leading to an increased insulin synthesis and secretion. Therefore, in light of available studies, AQP7 appears to be involved in the control of insulin release. However, further experiments using conditional *Aqp7* knockout mice could be useful to refine our understanding of the role of AQP7 in β-cells insulin secretion.

AQP12, an unorthodox AQP, was also found to be expressed in rat β-cells and rat RIN-m5F β-cell line [65]. However, the possible involvement of AQP12 in β-cell insulin release remains to be assessed.

Interestingly, due to the similarities in the *Aqp7*^-/-^ and *Aqp7*^+/+^ β-cells response to hypoosmolarity (rates and degrees of swelling), it seems reasonable to speculate at least one water-facilitated pathway to be present in addition to AQP7, such as AQP5 or AQP8 [60] or Na-K-2Cl [72] (also shown to transport water [73]), shown to be expressed as well in β-cells. However, further studies will be necessary to assess the function of AQP5 and AQP8, and of Na-K-2Cl, in β-cells response to hypoosmolarity.

### 2.3. Diabetes, Obesity and Metabolic Syndrome

Due to AQP7’s expression, especially in β-cells, AQP7 became a protein of interest in light of pathologies affecting endocrine pancreas, in particular diabetes, obesity and metabolic syndrome [11,12,13,14,74,75,76].

In humans, the *AQP7* gene is localized to a chromosomal region with reported linkage to type 2 diabetes [77] and metabolic syndrome [78]. Single nucleotide polymorphisms in the *AQP7* gene have been associated with obesity and/or type 2 diabetes in Caucasians [79,80] and with type 2 diabetes in the Chinese Han population [81]. Identified missense (R12C, V59L and G264V) and silent (A103A and G250G) *AQP7* gene mutations in a cohort of Japanese subjects were not linked to obesity or diabetes [82]. In a cohort of Caucasian subjects, a subject with the G264V mutation in the *AQP7* gene presented type 2 diabetes, overweight and extremely low glycerol levels [79]. Additional studies are required to deepen the current knowledge concerning the impact of *AQP7* loss-of-function mutations or single nucleotide polymorphism in diabetes, obesity and metabolic syndrome.

The phenotype of *Aqp7* knockout mice is characterized by adult-onset obesity and hyperglycemia [61,62,83]. In obese rats, sleeve gastrectomy modified several parameters of the pancreas or linked to the pancreatic function. Indeed, sleeve gastrectomy decreased β-cell apoptosis, pancreatic steatosis, insulinemia, fasting blood glycaemia and improved insulin sensitivity of the obese rats [65]. In obese rats presenting increased pancreatic AQP7 and AQP12 expression, sleeve gastrectomy significantly increased AQP7 expression, but not AQP12 expression [65]. The use of isolated β-cells and/or single cell transcriptome analysis could provide data to assess AQP12 expression in β-cells, as AQP12 is also expressed in the exocrine pancreas [84]. Finally, additional investigations are required to assess whether AQP7 and/or AQP12 could become suitable therapeutic targets for the treatment of obesity and/or type 2 diabetes.

Interestingly, a very recent study showed that epigenetic modification, i.e., the methylation of the *AQP7* gene promoter region in human hypertrophic white adipose tissue correlated with decreased *AQP7* expression [85].

## 3. Exocrine Pancreas

### 3.1. Morphology and AQPs Distribution

Exocrine pancreas accounts for circa 90% of total pancreatic cells and consists of acinar and ductal epithelial cells involved in pancreatic juice secretion, representing daily about 1 to 2 liters and consisting of fluid and enzymes. Pancreatic juice secretion is required for neutralization of stomach acid and proper food digestion. Pancreatic juice secretion is controlled by several neurotransmitters including secretin, cholecystokinin and acetylcholine [86].

*AQP1*, *AQP3*, *AQP4*, *AQP5*, *AQP8* and *AQP12* mRNA were found in the human exocrine pancreas [87,88]. However, only AQP1, AQP5 and AQP8 protein expression were detected [87,88]. AQP1 expression was localized to centroacinar cells (apical membrane), intercalated ductal cells (apical membrane), capillary endothelial cells, and pancreatic zymogen granule membrane [87,89,90]. AQP5 was localized to intercalated ductal cells [87], AQP8 was found solely in the apical membrane of acinar cells [91] and AQP12 localization remained to be assessed.

Rat exocrine pancreas expresses similar *Aqps* mRNA as the human pancreas, with the exception of *Aqp3* [87,88,92]. In rat, AQP1 is localized in intralobular and interlobular ductal cells (apical membrane, basolateral membrane, caveolae and vesicular structures) [93,94], acinar zymogen granules [89] endothelia cells [92]. AQP3 is expressed in acinar cells (membrane and intracellular structures) [48]. In addition, AQP5 is expressed in centroacinar cells (apical membrane) and intercalated ductal cells (apical membrane) [87], and AQP8 is expressed in acinar cells (apical membrane) [88]. Mouse exocrine pancreas was reported to express AQP1 in interlobular ductal cells (apical membrane) [87], AQP5 in interlobular, intralobular and intercalated ductal cells (apical membrane) [87] and AQP12 in acinar cells (intracellular structures; Table 3) [84].

### 3.2. Physiology and AQPs Functions

Pancreatic juice secretion involves two distinct molecular steps: A first step consisting into the secretion of an isotonic fluid by acinar cells, and a second step consisting in the secretion of most of the water with sodium, chloride and bicarbonate ions by ductal cells [95]. In this two steps process, acinar AQP8, (apically located), and both ductal AQP1 (apically and basolaterally located) and AQP5 (apically located) ensure transcellular water flow to the gland lumen [87]. So far, pancreatic fluid secretion was shown do not be altered in *Aqp1*, *Aqp5, Aqp8* or *Aqp12* knockout mice [96,97,98], despite the major involvement of AQP8 and AQP1 in acinar and ductal fluid secretion, respectively [92,93]. However, very recently, fluid secretion measurement performed on isolated pancreatic ducts from *Aqp1* knockout and in vivo MRI cholangiopancreatography assessing the rate of pancreatic fluid secretion supported the involvement of AQP1 in pancreatic ductal fluid and bicarbonate secretion [99]. Similar techniques could be used for further investigating the respective contribution of other AQPs to the pancreatic juice secretion process.

### 3.3. Pancreatic Diseases including Pancreatitis, Cystic Fibrosis and Cancer

AQPs have been involved in several pancreatic diseases including pancreatitis, cystic fibrosis and cancer [9,29,30,48].

In acute or chronic pancreatitis, considered as inflammatory syndromes, patients suffer from sudden disease onset characterized by abdominal pain, elevated serum digestive enzymes levels and abnormal abdominal imaging. In a rat model of acute pancreatitis and in liver *X receptor β* knockout mice model displaying exocrine pancreatic insufficiency, AQP1 expression was decreased [100,101]. Despite several studies, the role of AQP1 in pancreatitis has not been fully understood [9]. Pancreatitis can cause multiple organ failure including lung and colon displaying the altered expression of AQPs [102,103,104]. In a mouse model of autoimmune pancreatitis, it is worth pointing out that the cystic fibrosis transmembrane conductance regulator (CFTR) corrector C18 and the CFTR potentiator VX770 rescued CFTR expression, restored AQP5 expression and pancreatic fluid secretion, and eliminated tissue inflammation [105]. It would be worth assessing whether CFTR corrector and potentiator are also able to correct AQP1 expression in animal models of pancreatitis. In the future, treatment with CFTR correctors may offer an additional therapeutic tool to treat pancreatitis. In the context of gene therapy for pancreatic disorders such as pancreatitis [106], adenoviral vectors encoding AQP1 or AQP5 may represent additional tools [32,36,107].

In cystic fibrosis, a genetic disease resulting from *CFTR* mutation, patients experienced several manifestations including alteration of pancreatic juice secretion. CFTR is normally expressed in pancreatic ductal cells and contributes to pancreatic juice secretion [108]. Interestingly, *Cftr* knockout guinea-pigs and mice showed decreased AQP1 expression and pancreatic fluid secretion [99,109]. Nevertheless, the underlying causal mechanism remains to be assessed.

In cancer, a leading cause of death worldwide, AQPs have been shown to be involved in cancer cell migration, adhesion, growth, proliferation, invasion and metastasis, as well as drug resistance, angiogenesis and epithelial–mesenchymal transition [27,28,49,50,110,111]. Studies have explored the role of AQPs in pancreatic cancers, especially in pancreatic ductal adenocarcinoma (PDAC) representing the most common type and being the most aggressive and lethal malignancy, with a five-year survival of only 7%. Pancreatic cancer is anticipated to become the second leading cause of cancer-related death by 2030 [112]. Currently, surgery represents the only therapeutic option to cure PDAC cancer, but one needs to keep in mind, that only a small number of patients present with a resectable tumor at the time of diagnosis. Early PDAC diagnosis remains difficult due to the lack of distinct symptoms and the absence of specific clinical markers of early stages of PDAC. Following the use of PDAC cell lines, AQPs involvement was suggested in PDAC cell migration, cell proliferation and increased apoptosis [91,113,114]. However, the expression of AQPs has been poorly studied in PDAC [29,30]. A first study performed using PDAC from a small cohort of Caucasian patients showed a modification in AQP5 localization and expression in intercalated and intralobular ductal cells as compared to normal pancreatic tissues [48]. Indeed, AQP5 labeling was localized to the entire plasma membrane and intracellularly as opposed to the typical apical membrane in normal pancreatic tissue [48]. In addition AQP5 expression was higher in PDAC and correlated with the tumor differentiation status and aggressiveness [48]. Furthermore, the same PDAC also displayed modified AQP3 localization and expression as compared to normal pancreatic tissues [48]. Indeed, AQP3 labeling was localized to ductal cells (plasma membrane or intracellularly) in PDAC (but heterogeneous among PDAC samples) while nearly absent in normal pancreatic ductal cells [48]. In addition, AQP3 expression was inversely correlated with the tumor differentiation status and aggressiveness [48]. A second study using PDAC from a relative larger cohort of Chinese patients displayed higher AQP1 and AQP3 expression as compared to benign pancreatic tissue, and the expression level was inversely correlated with the tumor, node metastasis stage of the disease [115]. In addition, positive AQP1 and AQP3 expression were significantly associated with patients survival but represent poor-prognosis factors in PDAC [115]. Further studies using larger cohort of PDAC patients are required to assess further the beneficial value of using AQPs as markers of PDAC stages and prognosis, and to support the possible involvement of AQPs in epithelial–mesenchymal transition occurring in PDAC. Finally, additional studies will be required to decipher the molecular mechanisms underlying AQPs differential expression in PDAC, and the benefit of using appropriate therapeutic tools to target AQPs in PDAC [31,50,111].

## 4. AQPs as Useful Tools for Clinicians

### 4.1. AQPs as A Target for Therapy

Due to their involvement in various cellular physiological and pathophysiological processes, AQPs represent targets for drug, phyto-compounds, antibody immunotherapy and gene therapy.

A variety of small molecules targeting AQPs have been extensively studied [11,31,33,34]. Indeed, AQPs can be inhibited by metal ions, gold(III) compounds, metalloids, antidiabetic drugs and other drugs or phyto-compounds [11,31,33,34,116]. In addition, AQPs expression can be modulated by various phyto-compounds including flavonoids, curcuminoids, stilbenes, chalcones, isoflavonoids, triterpenes, monoterpenoids and capsininoids [116,117,118]. Moreover, antibodies could also represent a useful tool to target AQPs [34]. Indeed, gene therapy using adenoviral vectors encoding AQP has already been used to restore AQP function [32,36,107,119].

Although all these tools have not been tested yet for their potential beneficial effects for the treatment of pancreatic pathologies, some of the tools could be evaluated for the treatment of diabetes, pancreatitis, cystic fibrosis and cancer [11,13,14,31,33,74,116]. Due to epigenetic modifications of *AQPs* genes, tools controlling such processes might also be useful to modulate AQPs expression [85,120,121]. In the future, additional studies will determine if AQPs can become new useful therapeutic targets for the treatment of pancreatic pathologies.

### 4.2. AQPs as Biomarkers

Predictive and prognostic biomarkers are valuable tools in cancer diagnosis and therapy monitoring. A very recent study assessed the expression of AQPs on 106 PDAC resected tissues samples and showed higher expression of AQP1 and AQP3 in PDAC compared with peritumoral tissue and normal pancreatic tissue. Both AQP1 and AQP3 are proposed as diagnostic markers of PDAC and a predictive marker of poor prognosis in PDAC patients [115]. Another study, conducted on 35 PDAC patients suggests AQP3 to be related with late and more aggressive stages of PDAC while AQP5 is proposed as a potential histological marker for early stages of PDAC [48].

Another report suggested that AQP3 expression in PDAC is negatively regulated by miR-874, and resulted in the suppression of cell proliferation and increased cell apoptosis in PDAC cell lines [113]. Further investigation should focus on the prognostic value of AQP3 and/or miR-874 using a larger cohort of clinical samples.

Furthermore, it is speculated that AQPs regulation may be implicated in the ability of some cancer cells to respond to treatment. For example, AQP3 overexpression has been shown to contribute to chemo-resistance to arsenite in melanoma [122], as well as to cisplatin in gastric cancer [123]. In colorectal cancer cells, knockdown of *AQP5* sensitizes cells to 5-fluorouracil via inhibition of the Wnt-β-catenin signaling pathway [124]. Whether the AQPs expression and regulation contribute to PDAC response to treatment remains an open question that surely deserves further investigation.

## 5. Conclusions

AQPs are expressed in both endocrine and exocrine pancreas ensuring important physiological functions related to insulin secretion and pancreatic fluid secretion. Several AQPs present modified expression in various pancreatic pathological conditions, making them exciting potential new drug target for pancreatic disorders. Furthermore, several recent studies suggest involvement of various AQPs in pancreatic cancer development. However, further studies are needed to understand fully the role of AQPs in the development of pancreatic malignancies and to confirm their potential use as prognostic and/or predictive biomarkers in pancreatic cancer.

## Figures and Tables

**Table 1 ijms-20-05052-t001:** Mammalian aquaporin (AQP) classification.

Mammalian AQPs Classes	AQPs	Permeability
Classical AQPs	AQP0, AQP1, AQP2, AQP4, AQP5, AQP6, AQP8	Water
Aquaglyceroporins	AQP3, AQP7, AQP9, AQP10, AQP11	Water, glycerol, urea, small solutes
Unorthodox AQPs	AQP11, AQP12	Uncertain for AQP12
Peroxiporins	AQP1, AQP3, AQP5, AQP8, AQP9, AQP11	Hydrogen peroxide

**Table 2 ijms-20-05052-t002:** AQPs expression in endocrine pancreas.

AQP	Endocrine Pancreas
AQP5	β-cells (m)
AQP7	β-cells (m, r)
AQP8	β-cells (m)
AQP12	β-cells (r)

m—mouse; r—rat.

**Table 3 ijms-20-05052-t003:** AQPs expression in exocrine pancreas.

AQP	Exocrine Pancreas
AQP1	Acinar cells (r, h) Ductal cells (m, r, h), Endothelial cells (r, h)
AQP3	Acinar cells (h)
AQP5	Acinar cells (r) Ductal cells (m, r, h)
AQP8	Acinar cells (r, h)
AQP12	Acinar cells (m)

m—mouse; r—rat; h—human.

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
