# Peer review of "Aquaporins Involvement in Pancreas Physiology and in Pancreatic Diseases"

_ijms, 2019, doi:10.3390/ijms20205052_

Round 1

Reviewer 1 Report

Arsenijevic and colleagues present a timely and well-written review about the role of aquaporins in the physiology and pathophysiology of the endocrine and exocrine pancreas. Nonetheless, some specific points require to be amended.

Specific comments

Line 45 and Table 1: please change “perioxoporins” by “peroxoporins”. In addition to the mentioned AQPs, AQP11 has been very recently described as a peroxoporin (Bestetti S et al Redox Biol. 2019, PMID: 31546170). Line 89: the expression of aquaglyceroporin AQP7 has been also detected in rat β cell line RIN-m5F (Méndez-Giménez L et al. Int J Obes 2017, PMID: 28584298). Line 90 and Table 12: the superaquaporin AQP12 is also expressed in b-cells of rat Langerhans islets (Méndez-Giménez L et al. Int J Obes 2017, PMID: 28584298). Lines 138-156, in this paragraph the authors explain the role of AQP7 in insulin secretion in b-cells. The incretin hormone glucagon-like peptide-1 (GLP-1) is among the most widely studied modulators of β-cell function, with the incretin effect accounting for 70% of the insulin secretion after an OGTT. Interestingly, GLP-1 downregulates AQP7 in RIN-m5F β-cells with AQP7 expression being negatively associated with insulin release (Méndez-Giménez L et al. Int J Obes 2017, PMID: 28584298). In this regard, it seems plausible that the reduction of AQP7 induced by GLP-1 might result in intracellular glycerol accumulation, leading to an increased insulin synthesis and secretion. Lines 157-159, AQP7 not only participates in insulin secretion in β-cells, but also in the regulation of triacylglycerol accumulation. Aqp7-deficient mice exhibit increased intraislet glycerol and TG content (Matsumura K et al, Mol Cell Biol. 2007, PMID: 17576812). Interestingly the orexigenic and lipogenic horone ghrelin downregulates AQP7, leading to an increased cytosolic glycerol content that promotes triacylglycerol synthesis (Méndez-Giménez L et al. Int J Obes 2017, PMID: 28584298). According to the approved gene nomenclature, human gene symbols generally are italicized, with all letters capitalized, without greek symbols and without hyphens (i.e. AQP7 for the gene encoding aquaporin-7). Murine and rat gene symbols should be in italics with only the first letter capitalised, without greek symbols and without hyphens (i.e. Aqp7 for the gene encoding aquaporin-7). Human, rat and murine protein designations should be the same as the gene symbols except that all letters should be capitalized, in roman and with hyphens when necessary (i.e. AQP7). Please, change the human, rat and murine gene and protein symbols accordingly throughout the manuscript and figures. Please use SI units (e.g. mol/L). A space must be included between the number and units (e.g. 50 mmol/L) except from percentages (e.g. 10%).

Author Response

We thank the reviewers for their constructive comments and suggestions to improve the manuscript. The manuscript was revised accordingly.

Reviewer 1

-Line 45 and Table 1: please change “perioxoporins” by “peroxoporins”. In addition to the mentioned AQPs, AQP11 has been very recently described as a peroxoporin (Bestetti S et al Redox Biol. 2019, PMID: 31546170).

The remarks have been taken into account and introduced in line 46 and in table 1. Please note that we named aquaporins involved in H202 transport « peroxiporins »  in accordance with the current bibliography in the field.

 -Line 89: the expression of aquaglyceroporin AQP7 has been also detected in rat β cell line RIN-m5F (Méndez-Giménez L et al. Int J Obes 2017, PMID: 28584298).

Line 89 was updated accordingly.

- Line 90 and Table 12: the superaquaporin AQP12 is also expressed in b-cells of rat Langerhans islets (Méndez-Giménez L et al. Int J Obes 2017, PMID: 28584298).

Line 90 and Table 2 were updated accordingly.

-Lines 138-156, in this paragraph the authors explain the role of AQP7 in insulin secretion in b-cells. The incretin hormone glucagon-like peptide-1 (GLP-1) is among the most widely studied modulators of β-cell function, with the incretin effect accounting for 70% of the insulin secretion after an OGTT. Interestingly, GLP-1 downregulates AQP7 in RIN-m5F β-cells with AQP7 expression being negatively associated with insulin release (Méndez-Giménez L et al. Int J Obes 2017, PMID: 28584298). In this regard, it seems plausible that the reduction of AQP7 induced by GLP-1 might result in intracellular glycerol accumulation, leading to an increased insulin synthesis and secretion.

Thank you for this suggestion, the paragraph was modified according to the suggestion.

-Lines 157-159, AQP7 not only participates in insulin secretion in β-cells, but also in the regulation of triacylglycerol accumulation. Aqp7-deficient mice exhibit increased intraislet glycerol and TG content (Matsumura K et al, Mol Cell Biol. 2007, PMID: 17576812). Interestingly the orexigenic and lipogenic horone ghrelin downregulates AQP7, leading to an increased cytosolic glycerol content that promotes triacylglycerol synthesis (Méndez-Giménez L et al. Int J Obes 2017, PMID: 28584298).

The lines 157-159 were updated to include ghrelin influence on AQP7 expression and the references have been updated according to the comment.

-According to the approved gene nomenclature, human gene symbols generally are italicized, with all letters capitalized, without greek symbols and without hyphens (i.e. AQP7 for the gene encoding aquaporin-7). Murine and rat gene symbols should be in italics with only the first letter capitalised, without greek symbols and without hyphens (i.e. Aqp7 for the gene encoding aquaporin-7). Human, rat and murine protein designations should be the same as the gene symbols except that all letters should be capitalized, in roman and with hyphens when necessary (i.e. AQP7). Please, change the human, rat and murine gene and protein symbols accordingly throughout the manuscript and figures.

Human, rat and murine gene and protein symbols were changed throughout the manuscript and tables.

-Please use SI units (e.g. mol/L). A space must be included between the number and units (e.g. 50 mmol/L) except from percentages (e.g. 10%).

The SI units were modified throughout the manuscript.

Reviewer 2 Report

The review is well structured and covers the different aspects of the role of aquaporin in the pancreas and in relation to both the physiological aspects and the relationships with pancreatic diseases. The bibliography is complete and updated to the latest work on aquaporin. However, if the authors agree, I would suggest introducing aspects of aquaporin evolution as a basis for the different roles and functions performed and distribution at the introduction or discussion level.

Author Response

We thank the reviewers for their constructive comments and suggestions to improve the manuscript. The manuscript was revised accordingly.

Reviewer 2

The review is well structured and covers the different aspects of the role of aquaporin in the pancreas and in relation to both the physiological aspects and the relationships with pancreatic diseases. The bibliography is complete and updated to the latest work on aquaporin. However, if the authors agree, I would suggest introducing aspects of aquaporin evolution as a basis for the different roles and functions performed and distribution at the introduction or discussion level.

We added to the introduction a paragraph covering aquaporin evolution as a basis for the different roles, functions and tissue distribution, while remaining within the scope of the article.